# Retrospective Detection and Complete Genomic Sequencing of *Canine morbillivirus* in Eurasian Otter (*Lutra lutra*) Using Nanopore Technology

**DOI:** 10.3390/v14071433

**Published:** 2022-06-29

**Authors:** Zsófia Lanszki, József Lanszki, Gábor Endre Tóth, Safia Zeghbib, Ferenc Jakab, Gábor Kemenesi

**Affiliations:** 1National Laboratory of Virology, Szentágothai Research Centre, University of Pécs, 7624 Pécs, Hungary; lanszkizsofi@gmail.com (Z.L.); toth.gabor.endre@gmail.com (G.E.T.); zeghbib.safia@gmail.com (S.Z.); jakab.ferenc@pte.hu (F.J.); 2Faculty of Sciences, Institute of Biology, University of Pécs, 7624 Pécs, Hungary; 3Department of Nature Conservation, Hungarian University of Agriculture and Life Sciences, 7400 Kaposvár, Hungary; lanszkij@gmail.com

**Keywords:** *Mustelidae*, NGS, third generation sequencing, conservation biology, MinION, enzootic

## Abstract

The Eurasian otter (*Lutra lutra*) is a piscivorous apex predator in aquatic habitats, and a flagship species of conservation biology throughout Europe. Despite the wide distribution and ecological relevance of the species, there is a considerable lack of knowledge regarding its virological and veterinary health context, especially in Central Europe. *Canine morbillivirus* (Canine distemper virus (CDV)) is a highly contagious viral agent of the family *Paramyxoviridae* with high epizootic potential and veterinary health impact. CDV is present worldwide among a wide range of animals; wild carnivores are at particular risk. As part of a retrospective study, lung-tissue samples (*n* = 339) from Eurasian otters were collected between 2000 and 2021 throughout Hungary. The samples were screened for CDV using a real-time RT-PCR method. Two specimens proved positive for CDV RNA. In one sample, the complete viral genome was sequenced using a novel, pan-genotype CDV-specific amplicon-based sequencing method with Oxford Nanopore sequencing technology. Both viral sequences were grouped to a European lineage based on the hemagglutinin-gene phylogenetic classification. In this article, we present the feasibility of road-killed animal samples for understanding the long-term dynamics of CDV among wildlife and provide novel virological sequence data to better understand CDV circulation and evolution.

## 1. Introduction

The Eurasian otter (*Lutra lutra*) is a flagship species of nature conservation efforts throughout Europe, and it is a widely distributed piscivorous carnivore in Eurasia and portions of North Africa [1,2]. The otter is a characteristic apex predator in aquatic food chains [2]. It inhabits a wide variety of natural habitats (e.g., rivers, small waterflows, lakes and marshlands) and human-altered areas (fishponds, water reservoirs and recreational lakes); it is mainly solitary, secretive, and nocturnal [1,2]. Currently, it is a near-threatened species on the IUCN Red List [3], and it is listed as an animal species of European Community importance (EEA, 2009). The reason for its priority protection is the vulnerability of its population [1,2]. Recent decades have seen an increase in otter populations in some areas of Europe [4,5], which implies that otters are more likely to encounter humans, domestic dogs, and other carnivores. In Hungary, the otter is a widespread, strictly protected species with stable, interconnected populations [6,7].

*Canine morbillivirus* (canine distemper virus (CDV)) is a single-stranded negative-sense RNA virus that belongs to the *Paramyxoviridae* family in the *Morbillivirus* genus [8,9,10]. Several distinct genotypes are known and classified according to different hosts and geographical areas, based on the nucleotide sequence analysis of the hemagglutinin (H) gene [11,12]. Functionally, the hemagglutinin protein is responsible for virus’ attachment and fusion to host cells. It is widely used to assess genetic relatedness among different CDV strains [13,14,15]. Canine distemper virus is a significant viral pathogen, affecting wildlife and domestic animals [16,17]. It has been reported among multiple species of wild carnivores from the *Mustelidae* family, such as the stone marten (*Martes foina*), the pine marten (*Martes martes*), the Eurasian badger (*Meles meles*) [17,18,19,20,21,22], and otter species [22,23,24,25,26,27,28,29,30].

CDV threatens a wide range of endangered animal populations. The disease’s progression is usually coupled with high mortality, virulence, and frequent cross-species transmission events. Therefore, apart from a general veterinary health problem, it is also a significant conservation threat to endangered species worldwide [16,31,32,33]. Infectious diseases are critical limitation factors regarding the population size and dispersal of wildlife species. Therefore, there has been a growing interest, during the last decades, in emerging infectious diseases in wildlife [34,35]. In several situations, it was clearly shown that most dispersed and small populations of endangered wild animals are more prone to extinction due to stochastic events, such as disease outbreaks [36,37,38]. Clearly, disease monitoring is deemed important in the conservation of rare species. However, the examination of wild animals, especially carnivores, is often more difficult than that of domestic or zoo animals due to ethical reasons and sampling or detection difficulties. Population-size determination, morbidity, mortality estimation, and the early detection of disease outbreaks are highly challenging, specifically, among wildlife species [39]. The results of post-mortem studies may provide novel and relevant data for rare and hidden species, including the prevalence and genetic characteristics of certain infectious diseases [26,40].

In addition to the detection of pathogens, next-generation sequencing (NGS) technologies are increasingly used in microbiology laboratories for the in-depth characterization of pathogens [41]. NGS can be optimized for rapid sample preparation and real-time sequence analysis, as well as the rapid identification of certain pathogens [42,43,44,45,46,47]. Amplicon-based NGS sequencing is an increasingly preferred method in the rapid detection and genomic characterization of specific pathogens, with high specificity and sensitivity [48,49,50,51,52].

In terms of virological examinations, the Eurasian otter is a neglected predator species; it is crucial to understand the dynamics, risks, and evolution of the most common viral diseases regarding of this species. Understanding CDV’s long-term epidemiology in the Eurasian otter is of utmost importance from a conservation perspective, and the presentation of novel sequence data is also highly relevant for better understanding CDV’s evolution. In this paper, we present the results of a retrospective surveillance study spanning 21 years to detect CDV in road-killed Eurasian otter samples. The reliability of such samples for virological studies using a highly specific NGS method for the rapid genomic characterization of CDV-positive samples is presented.

## 2. Materials and Methods

### 2.1. Retrospective Collection of Eurasian Otter Samples

We collected Eurasian otter carcasses between 2000 and 2021 in Hungary. These animals were primarily (90%) road-killed individuals, whilst the remaining animals were found dead at their natural habitats. Animal collection localities cover two habitat types (stagnant waters or watercourses) and highlight the distribution of these animals within the country [6]. The animal carcasses were collected by the staff of the ten National Park Directorates and stored at −20 °C until processing. The post-mortem examination was carried out and tissue samples of different organs were stored at −20 °C by the Carnivore Ecology Research Group, Kaposvár University [53,54,55], with permission from the competent authorities. A total of 339 lung tissue samples were collected from the carcasses using general dissection procedures.

### 2.2. Nucleic Acid Extraction and PCR Reactions

Approximately five grams of lung tissues were homogenized in 500 µL of phosphate-buffered saline (PBS), using the Bertin Minilys machine at maximum speed for 3 min, supplemented with two glass beads per sample to facilitate tissue disruption. Following brief centrifugation, 100 µL of the supernatant was used for RNA extraction using the Monarch total RNA miniprep kit (NEB, Ipswich, MA, USA). The nucleic acid samples were tested using a CDV-specific real-time PCR [10]. All PCRs were performed using the OneTaq^®^ One-Step RT-PCR Kit (NEB, Ipswich, MA, USA) in full compliance to the manufacturer’s recommendations. RT-PCRs were performed immediately following RNA extraction without freeze-thawing the nucleic acid to avoid possible RNA degradation for improved output in complete genomic sequencing activities. The RT-PCR standard curve was generated by serial 10-fold dilutions of a corresponding CDV amplicon with a known copy number in a range of 1 × 10^10^ to 1 × 10^1^. These dilutions were measured in triplicate, and the measured results were used to construct the standard curve, which was subsequently used to determine the copy number from threshold cycle values (Ct) of the samples.

### 2.3. Sanger Sequencing and MinION Sequencing

In the cases in which the virus titer was too low for complete viral genome sequencing, we applied a specific PCRs reaction, targeting only the hemagglutinin (H) gene of CDV with previously published primer sets [56]. Sanger sequencing was performed by an external service provider (Eurofins Genomics, Germany).

The complete genome sequencing was performed with MinION nanopore sequencing technology (Oxford Nanopore Technologies, Oxford, UK). Previously, we developed and published an amplicon-based sequencing method in reference to CDV genome sequencing [51]. MinION library preparation, sequencing, and data analysis were performed in an identical manner. The detailed protocol is available at our laboratory protocols.io page [57].

### 2.4. Phylogenetic Analysis

Two datasets were used for phylogenetic tree analysis comprising 180 and 843 complete genomic and complete hemagglutinin gene sequences, respectively. Sequences were aligned in MAFFT webserver using default parameters. Subsequently, in IQ-TREE webserver, both best substitution model selection and maximum-likelihood phylogenetic tree reconstruction were performed with ultrafast bootstrapping. Phocine distemper virus (PDV) was used as an outgroup for both phylogenies.

## 3. Results and Discussion

### 3.1. RT-PCR Screening

Eurasian otter samples were collected from all nineteen counties throughout Hungary. From these, canine distemper virus RNA was identified by real-time RT-PCR screening in 2/339 samples. Both originated from western Hungary; one was collected in 2006 and the other in 2010. The first infected animal was a young (4–5 months old) male in poor body condition (K = 0.80; for condition index or K) [2] and was found to be deceased due to natural infection, on the edge of a marshland (Kis-Balaton). The second was an adult male in good body condition (K = 1.19), found as road-kill near a river (Rába) [6]. Consequently, only 2 of the 339 samples were found to be CDV-positive, although this does not exclude the possibility of the presence of CDV in the other samples, since we had no data on the viral RNA degradation in these samples. However, the objective of this study was not the assessment of viral prevalence, but to demonstrate the feasibility of such samples in viral genomic surveillance.

Although the Eurasian otter is a well-studied “key species” of aquatic habitats in ecological and zoological terms [2], there is a considerable lack of knowledge of the microbiological context across its distribution range. Aleutian mink disease parvovirus, carnivore protoparvovirus 1, feline panleukopenia virus, and canine adenovirus type 1 were previously detected in the species [58,59,60]. Regarding the *Morbillivirus* virus family, *Dolphin Morbillivirus* was detected in Eurasian otters [61]. Nevertheless, the presence of canine distemper virus was only described based on the histology among European otters [25,29]. CDV infections were reported among the members of the family *Mustelidae*, and the virus has been detected in several different otter species. Under zoo conditions in Belgium, CDV was observed in littermates of the Asian small-clawed otter (*Aonyx cinereus*), based on histopathology and direct immunofluorescence [28]. CDV was detected in a sea otter (*Enhydra lutris*) population using immunohistochemistry, RT-PCR, genetic sequencing, virus isolation, and serology in upstate Washington, USA [26]. North American river otters (*Lontra canadensis*) were seropositive against canine distemper virus in the northern and eastern regions of the USA, implying its circulation among these animals [27]. Seemingly, CDV infections are present in a wide range of species within the *Mustelidae* family. Considering their relevance regarding their conservation biology, vaccination may provide a solution for avoiding epizootic events within otter populations. The vaccination of otters is not without precedent; it is reported and evaluated in multiple publications [62,63].

### 3.2. Sequencing Results

Based on the real-time RT-PCR results, the viral genomic copy numbers were calculated as follows: ~1,021,000 copies/μL of the sample from 2006 and <10 copies/μL from the sample from 2010. The complete genome nucleotide sequence was determined for the sample from 2006 using Oxford Nanopore sequencing technology (Figure 1). To our knowledge, these are the first two CDV genome sequences from the Eurasian otter. The full-length hemagglutinin (H) nucleotide sequence of the other sample was obtained with specific PCRs and Sanger sequencing. These sequence data were submitted to the GenBank databases (accession numbers OM811640 and OM811639).

We emphasize the importance of providing novel sequence data in accordance with recent studies, in which the complete genomic sequence data of CDV are increasingly reported. This will greatly facilitate a better understanding of both the evolution of CDV and epizootic patterns in the near future [47,64,65,66,67]. In our study, we used a recently published pan-genotype CDV-specific amplicon-based sequencing method developed for the Oxford Nanopore Technologies platform [51]. A key advantage regarding this method is its ability to sequence the entire genome of CDV quickly and efficiently with multiplexed amplicons, without the necessity for in vitro isolation procedures.

### 3.3. Phylogenetic Analysis

Based on the phylogenetic analysis of the hemagglutinin gene sequences, the two CDV sequences in our study belong to the European lineage. The complete-genomic-sequence-based analysis confirmed this observation (Figure 2). Both sequences clustered in a separate cluster among the other European sequences, separately from other clusters described for the other species, such as foxes. More studies are needed to reveal the presence and understand the risk of cross-species transmission events between otters and other carnivores, if indeed there are any. Based on our data, we demonstrate the presence of a separate strain, which was identified in two different years in the same region in Hungary. Increased efforts and extended research are needed to better understand the background of this genetic separation.

In Hungary, three different CDV genotypes were described, based on the H gene nucleotide sequence to date. In the last two decades, European, Arctic-like, and European wildlife lineage were detected among dogs. In addition to dogs, CDV infection was detected in wild carnivores, such as the fox (*Vulpes vulpes*), raccoon (*Procyon lotor*), and ferret (*Mustela putorius furo*) [51,68,69,70].

The European lineage of CDV was detected in many wild-animal species originating in various European countries. This lineage was frequently observed among red foxes and badgers in several countries and across multiple years, such as Germany in 2008 [56] and Italy between 2006 and 2009 [71,72]. Moreover, in Switzerland, numerous wild carnivores, including red foxes, Eurasian badgers, stone and pine martens, Eurasian lynx (*Lynx lynx*), and domestic dogs, were found with CDV lesions between 2009 and 2010 [19]. In Denmark, a major outbreak of CDV was detected in American minks (*Neovison vison*) originating on mink farms, as well as a high number of wild species, such as the red fox, raccoon dog (*Nyctereutes procyonoides*), and European polecat (*Mustela putorius*) between 2012 and 2013 [73]. Recently, the European lineage of CDV was detected in Germany among raccoons (*Procyon lotor*) from 2015, in red foxes from 2016 [74], and in Northern Italy in red foxes, badgers, and stone martens between 2018 and 2019 [75]. In addition to these studies, numerous red foxes were detected in Hungary in 2021, in association with a possible countrywide epizootic event [51]. Apparently, CDV is a widespread and, at the same time, fairly investigated pathogen among wild carnivores throughout Europe. We demonstrated the reliability of road-killed animal samples for retrospective virological examination regarding CDV genomic patterns, prevalence, and host range. The growing number of genomic data may significantly aid in comprehending and predicting future epizootic events.

## 4. Conclusions

Large sample sizes and retrospective studies are suitable for molecular biological examination and lead to a better understanding of CDV events occurring decades in the past. The examination of road-killed animals facilitates disease observation and surveillance among rare animal species, which includes the detection and genome sequencing of the virus, as it demonstrated in this study. The presence of CDV in otters draws attention to the potential threat to the population of this predatory species.

## Figures and Tables

**Figure 1 viruses-14-01433-f001:**
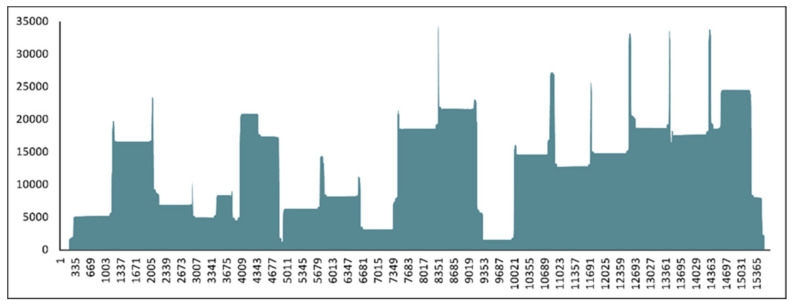
Visualization of sequencing coverage via the amplicon-based sequencing method for the sample from 2006, with all primer sets (1000 and 2000 bp). The horizontal scale represents the genomic position, while the vertical scale displays the coverage values of the sequencing reaction.

**Figure 2 viruses-14-01433-f002:**
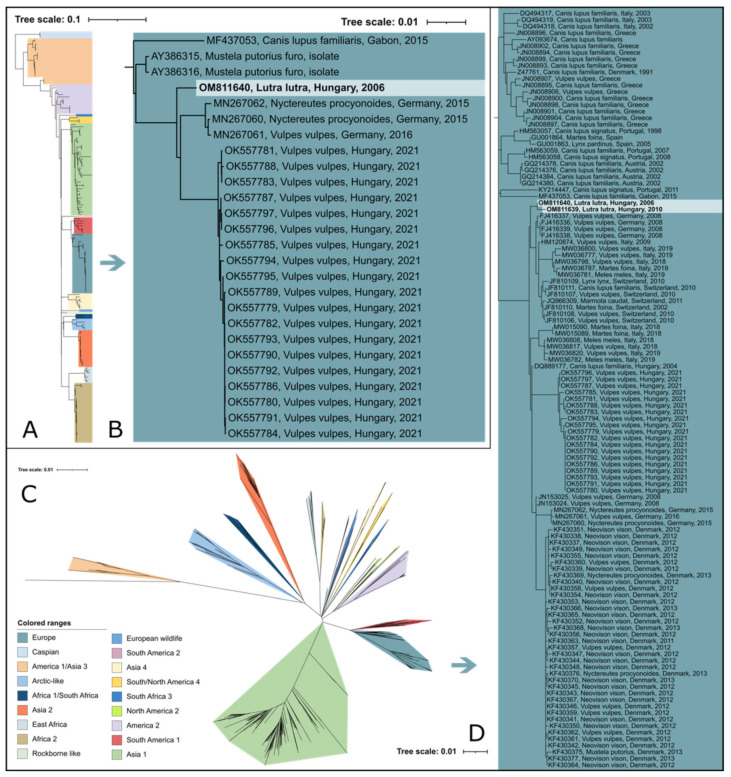
(**A**) Maximum-likelihood phylogenetic tree based on 180 CDV complete-genome nucleotide sequences. (**B**) The European lineage of interest is highlighted in blue color. (**C**) Maximum-likelihood phylogenetic tree based on 843 complete hemagglutinin (H) nucleotide sequences. (**D**) The European lineage of interest is highlighted in blue. Phocine distemper virus (PDV) (GenBank accession number: KY629928) was used as an outgroup to root both phylogenetic trees.

## Data Availability

Not applicable.

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
