# Peer review of "Retrospective Detection and Complete Genomic Sequencing of Canine morbillivirus in Eurasian Otter (Lutra lutra) Using Nanopore Technology"

_viruses, 2022, doi:10.3390/v14071433_

Round 1

Reviewer 1 Report

The authors have screened 339 otters found dead in Hungary in the period 2000-2021 by RT-PCR for the presence of canine distemper virus genome. They found two positive samples and provide one partial and one full genome sequence.

In my opinion, the manuscript lacks novelty that would justify a full paper in Viruses:

1. The surveillance of dead otters has already been published (refs 53-55)

2. The CDV sequencing method using nanopore technology has already been published (ref 51)

3. CDV infection of otters has been described before

4. The authors found only 2/339 animals positive, suggesting that CDV is not a huge clinical problem in otters in Hungary

5. Although I support the fact that full-length sequences are important for phylogeny, the full genome described here is from 2006 (i.e. not recent) and is similar to other previously published sequences in the genotype of European CDV strains

In my opinion this should not be a full paper, but a submission to a journal like Microbiology Resource Announcements.

Author Response

Responses to the Reviewer 1 comments

Thank you for your helpful comments on our manuscript. We appreciate the suggested modifications and have revised the manuscript accordingly and its final version is enclosed. We hope all your concerns are now fully addressed.

Reviewer 1

The authors have screened 339 otters found dead in Hungary in the period 2000-2021 by RT-PCR for the presence of canine distemper virus genome. They found two positive samples and provide one partial and one full genome sequence.

In my opinion, the manuscript lacks novelty that would justify a full paper in Viruses:

  1. The surveillance of dead otters has already been published (refs 53-55)

Response: In our opinion this study is not simply a surveillance study. As we mention in the manuscript, we intended to present the eligibility of retrospective sample collections for virus surveillance studies to better understand viral dynamics in rare animal species (such as the Eurasian otter), where direct/active sampling is not performable. We demonstrated the feasibility of such samples, even for obtaining complete genomic information of an RNA virus. We believe this work will encourage other researchers to use already existing sample banks for similar purposes. Such work is not without example, there are already published works, where authors are discussing the feasibility of road-killed animals for microbiological research purposes (such as: Szekeres S, Docters van Leeuwen A, Tóth E, Majoros G, Sprong H, Földvári G. Road-killed mammals provide insight into tick-borne bacterial pathogen communities within urban habitats. Transbound Emerg Dis. 2019 Jan;66(1):277-286. doi: 10.1111/tbed.13019. Epub 2018 Oct 11. PMID: 30230270.).

Summarizing our achievements, we presented the feasibility of long-storage retrospective samples to obtain complete viral genomic sequence with a third-generation sequencing technology. We demonstrated this workflow on Eurasian otter samples, giving a first-hand example to investigate rare animal species with this approach.

  1. The CDV sequencing method using nanopore technology has already been published (ref 51)

Response: The complete genome sequencing method was recently developed in our lab in red fox (Vulpes vulpes), we started using the method beyond the description. The aim of the present study was to detect CDV from Eurasian otter (Lutra lutra) and complete genome sequencing using this new method – from long-storage samples. That is one of our main achievements here, presenting the reliability of this method in retrospective studies. Amplicon-based sequencing is a powerful tool to obtain high-quality sequence data from low-quality samples.

  1. CDV infection of otters has been described before

Response: Indeed, CDV has already been detected with RT-PCR in other otter species, but only symptoms and histology have been described from the Eurasian otter (Lutra lutra) species. A total of 13 otter species live around the world. As far as we know, these are the first PCR positive cases as well as these are the first CDV genome sequences from this species. We hope that after our retrospective detection and genome sequencing results, other virological and population genetic research groups will also study the species to explore the effects of CDV on the otter population deeper.

We added more clarification in the manuscript text:

Lines 144-146. Nevertheless, the presence of Canine distemper virus was only described based on histology among European otters [25,29]

This information has been added to the text:

“To our knowledge, these are the first two CDV genome sequences from Eurasian otter.” (Lines 166-167)

  1. The authors found only 2/339 animals positive, suggesting that CDV is not a huge clinical problem in otters in Hungary

Response: Based on our results, we cannot draw conclusions about the impact of CDV on Eurasian otters. This study demonstrated the feasibility of long-storage samples to conduct viral genomic surveillance. Considering the randomity of samples this kind of passive sampling strategy is not feasible to draw conclusions on prevalence or impact. In our study, little is also of value since we were mostly interested in the presence of CDV in this species and providing viral genomic data with state-of-the-art method. The exact number of the mortality due to this disease is largely invisible and undetectable to us, because of the hiding lifestyle of the otter.

However, the otter population in Hungary is special in Europe, because there has been no decline in the population in the previous decades. The results obtained refer to a stable otter population on an ongoing basis and may serve as a basis for future comparison with similar studies in other Eurasian areas.

  1. Although I support the fact that full-length sequences are important for phylogeny, the full genome described here is from 2006 (i.e. not recent) and is similar to other previously published sequences in the genotype of European CDV strains

Response: To better understand CDV evolution, complete genomes can provide important information. Depending on the question, the temporality of the samples can be relevant is different ways. Since there is a significant lack of knowledge about the genomic evolution of CDV, our work has an added value by providing a demonstrative work on a novel concept to obtain more sequence data from already existing sample banks.

In my opinion this should not be a full paper, but a submission to a journal like Microbiology Resource Announcements.

Reviewer 2 Report

Thank you for this manuscript. It is well presented and easy to read and comprehend. It is also nice to see that some progress is made in detecting CDV in retrospective samples. I would however suggest the mention of CDVs ease in degrading in certain environmental situations being a single stranded RNA virus and the difficulty in identifying it in samples of "low" quality. I also suggest mentioning that even though only 2 of the 339 samples were found to be CDV positive it does not exclude the possibility of CDV being present in the other samples.

Author Response

Responses to the Reviewer 2 comments

Thank you for your helpful comments. We appreciate the suggested modification and have revised the manuscript accordingly.

Reviewer 2

Thank you for this manuscript. It is well presented and easy to read and comprehend. It is also nice to see that some progress is made in detecting CDV in retrospective samples. I would however suggest the mention of CDVs ease in degrading in certain environmental situations being a single stranded RNA virus and the difficulty in identifying it in samples of "low" quality. I also suggest mentioning that even though only 2 of the 339 samples were found to be CDV positive it does not exclude the possibility of CDV being present in the other samples.

We would like to thank you for your helpful comments on our manuscript. Indeed, RNA degradation may be considered during the collection of the dead animal (time between death and collection), dissection and examination of the samples, we have supplemented the text with this information.

Considering the possible virus degradation during collection and dissection procedures, the randomity of samples this this study design is not suitable to assess prevalence of the virus. Although that was not our main objective and by providing high-quality viral genomic data, we demonstrated the feasibility of this approach in viral genomic surveillance.

We added a sentence to the manuscript, referring to the possible viral degradation:

“Consequently, only 2 of the 339 samples were found to be CDV positive, but it does not exclude the possibility of CDV present in the other samples, since we have no data on viral RNA degradation in these samples. However, the objective of this study was not the assessment of viral prevalence, but to demonstrate the feasibility of such samples in viral genomic surveillance.” (Lines 138-142)

Reviewer 3 Report

In this study, the authors conducted a retrospective study in animal road-killed or elsewhere, focusing on Eurasian otters and CDV. And the authors indicated that the full-length genome could be determined from CDV-positive sample without virus isolation by using an amplicon-based sequencing method.

The study provides an important contribution to viral full-length genome analysis on retrospective study of wildlife. Because the number of CDV full-length genomes have been increasing in recent years, and their importance of the genome sequencing is increased in CDV epidemiological study.

The results are clearly presented and I have nothing point out.

Author Response

We are grateful for the honourable appreciation of our work and for the work during the revision process. We made some minor modifications and clarifications, based on the other reviewers reports. Thank you

Round 2

Reviewer 1 Report

I agree to disagree with the authors of the manuscript, and after reading their rebuttal to my comments still feel that the data do not justify a full paper in Viruses. However, since the other two reviewers were more positive than me, I do not intend to block publication.